# Resistance Training before, during, and after COVID-19 Infection: What Have We Learned So Far?

**DOI:** 10.3390/ijerph19106323

**Published:** 2022-05-23

**Authors:** Paulo Gentil, Claudio Andre Barbosa de Lira, Carlos Alexandre Vieira, Rodrigo Ramirez-Campillo, Amir Hossein Haghighi, Filipe Manuel Clemente, Daniel Souza

**Affiliations:** 1College of Physical Education and Dance, Federal University of Goias, Goiânia 74690-900, Brazil; andre.claudio@gmail.com (C.A.B.d.L.); vieiraca11@gmail.com (C.A.V.); daniel_souza86@hotmail.com (D.S.); 2Hypertension League Federal University of Goias, Goiânia 74605-050, Brazil; 3Instituto VIDA, Brasilia 70.000, Brazil; 4Exercise and Rehabilitation Sciences Laboratory, School of Physical Therapy, Faculty of Rehabilitation Sciences, Universidad Andres Bello, Santiago 7591538, Chile; rodrigo.ramirez@unab.cl; 5Department of Exercise Physiology, Faculty of Sport Sciences, Hakim Sabzevari University, Sabzevar 9617976487, Iran; ah.haghighi@hsu.ac.ir; 6Escola Superior de Desporto e Lazer, Instituto Politécnico de Viana do Castelo, 4900-347 Viana do Castelo, Portugal; filipe.clemente5@gmail.com; 7Research Center in Sports Performance, Recreation, Innovation and Technology (SPRINT), 4960-320 Melgaço, Portugal; 8Instituto de Telecomunicações, Delegação da Covilhã, 1049-001 Lisboa, Portugal

**Keywords:** human physical conditioning, resistance training, coronavirus, muscle strength, musculoskeletal and neural physiological phenomena

## Abstract

At the end of 2019, a severe acute respiratory syndrome caused by SARS-CoV-2 started a pandemic, leading to millions of deaths and many important political and social changes. Even in the absence of contamination, the mobility reduction, social distancing and closing of exercise facilities negatively affected physical activity and conditioning, which is associated with muscle atrophy, loss of muscle strength, and reductions in functional capacity. In cases of infection, it has been shown that increased physical capacity is associated with decreased hospitalization and mortality risk. Although millions of people have died from COVID-19, most contaminated individuals survived the infection, but carried different sequelae, such as the severe loss of physical function and a reduced quality of life. Among different physical exercise models that might help to prevent and treat COVID-19-related conditions, resistance training (RT) might be particularly relevant. Among its benefits, RT can be adapted to be performed in many different situations, even with limited space and equipment, and is easily adapted to an individual’s characteristics and health status. The current narrative review aims to provide insights into how RT can be used in different scenarios to counteract the negative effects of COVID-19. By doing this, the authors expect to provide insights to help deal with the current pandemic and similar events the world may face in the future.

## 1. Introduction

In December of 2019, there was an outbreak of a severe acute respiratory syndrome caused by a new coronavirus (SARS-CoV-2). The virus was first noticed in China and rapidly spread across the country and then across the world [1]. As a consequence, many authorities imposed extreme measures such as quarantines, social distancing, and isolation [2]. Measures included banning sports competition and closing exercise facilities, such as health clubs, gyms, and sport courts [3,4]. These restrictions were accompanied by a reduction in mobility due to public transportation and gathering restrictions, working from home adoption, and school closures. These measures had a negative impact on physical activity levels [5] and decreased the involvement with muscle strengthening exercises [6], which might induce muscle atrophy, loss of muscle strength, and reductions in neuro and mechanical abilities [7,8].

Even when exercise facilities were reopened, many regulations were sustained, such as social distancing, limited gathering, the use of protective masks, and hygiene measures [9,10], which, along with the fear of contamination, might preclude regular exercise performance. Therefore, it is important to propose solutions to stimulate physical activity performance, especially considering that physical inactivity [11,12,13,14,15] and low physical capacity [16,17,18] are associated with worse outcomes and an increased mortality risk in cases of infection [19,20,21].

Moreover, although COVID-19 is commonly associated with the respiratory system, it is a multisystem disease [22]. Coronaviruses may also induce neurological damage by invading the central nervous system and result in severe muscle pain [23] and loss of muscle strength [24]. Although COVID-19 has relevant morbidity for up to 6 months [25], COVID-19 survivors might develop psychological, physical and cognitive impairments that require rehabilitation and medical care for more than 12 months [26,27]. Among the secondary consequences of the disease and its treatment, physical function is unlikely to recover to normal values spontaneously, even under nutritional and physical exercise counselling [28]. Therefore, specific prescriptions are needed.

Considering physical exercise models that might help to prevent and treat the different consequences of COVID-19, resistance training (RT) might be particularly relevant. RT has been consistently used to increase muscle mass and strength in many different populations, being considered an essential part of a physical exercise program aiming to improve or restore physical functioning [29,30,31]. Its benefits, largely mediated by strength gains, culminate in reductions in mortality rates in different populations [32,33,34,35,36]. Therefore, this narrative review aimed to provide insights into how RT can be used in different scenarios to counteract the negative effects of COVID-19. By doing this, the authors expect to provide insights to help deal with the current pandemic and provide information in case the world has to deal with similar events in the future.

## 2. Before: Considerations for Preventing COVID-19 Complications

Although it is not possible to attribute a direct cause–effect relationship between RT practice and the mortality risk during the COVID-19 pandemic, current evidence suggests that it might be important to perform RT to improve general health and promote a better prognosis in cases of contamination [11,12,13,14,15,16,17,18,37]. Physical inactivity [11,12,13,14,15] and low muscle strength have been associated with an increased risk of hospitalization and death [16,17,18,37]. The importance of muscle strength should not be underestimated, since it may explain the protective effect of physical activity against COVID-19 hospitalization [17].

Moreover, RT can modulate important risk factors associated with increased morbidity and mortality due to COVID-19, such as high blood glucose, arterial hypertension, obesity, and dyslipidemia [38,39,40,41,42]. In this regard, previous evidence shows that RT can help to control blood pressure [43,44], blood glucose [45], body weight [46], and blood lipids [47]; therefore, it can mitigate complications in cases of contamination.

Another possible benefit of RT is its impact on the immune system. Physical exercise has been consistently shown to modulate immune function [48,49,50,51]. Higher levels of physical activity [52,53,54] and fitness [54] decrease the risk of respiratory symptoms and illness. In this regard, people who carry out strength and power activities [55,56,57,58] usually have a better immunological profile than people who perform long-duration aerobic activities [59], which might be a positive point for RT [60,61]. Strategies for RT prescription for improving or maintaining immune function involve using a low exercise volume (4–6 exercises, with 1–2 sets per exercise), avoiding metabolic stress (perform ≤ 6 repetitions and ≥2 min of rest between sets and exercises) and preferring exercising during the afternoon/evening [60].

Based on the aforementioned points, it is important to promote muscle-strengthening activities during the current and future pandemics [9,61], especially for those at higher risk of frailty, such as the elderly and people with chronic diseases [62,63]. Although many people might feel unsafe when practicing RT in exercise facilities, there are many possible measures that could be adopted [9], and previous studies have shown their relevance in controlling the contamination risk [64,65].

If one decides to avoid exercise facilities, RT can be adapted to be performed in many different situations, even with limited space and equipment, and it can easily be adapted to an individual’s characteristics and health status [61]. For example, previous studies have shown that bodyweight exercises [66,67,68], stationary bike training [69], plyometric training [70], elastic band training [71,72,73], and even exercises with no external load [74,75,76] promote similar responses to traditional RT. These exercises might be performed as basic multi-joint exercises (i.e., squats, pushups, pullups, rows, etc.) as this has been shown to be sufficient to promote gains in muscle strength and size in most muscles involved [77,78,79,80,81]; the addition of isolated exercises, in general, does not seem to bring benefits [80,82,83]. This allows the possibility to exercise at parks, outdoors, and even at home, and still obtain relevant results [63,84]. Additionally, these training approaches can be adjusted to intensify or decrease the intensity of the practice, which may be interesting for those who want to vary their training stimulus across long-term periods of gym avoidance.

Considering that the respiratory system of infected people is the main source of SARS-CoV-2 contamination, one of the most frequently recommended safety measures is to use protective masks [85,86]. However, it is important to consider that the use of any type of mask reduces air flow to the lungs and might increase respiratory stress, leading to dizziness, shortness of breath, and a decrease in performance [10,87,88,89]. Although this might be alleviated by familiarization [90], it might be recommendable to control respiratory responses during exercise. Here, RT might be particularly advantageous due to the less pronounced cardiorespiratory demands in comparison to aerobic exercise [91,92,93].

A previous study suggested that wearing face masks (surgical or FFP2) during RT resulted in similar strength performances and physiological responses to training with no mask when exercise was not performed to muscle failure [94]. Rosa et al. reported that the effects of FFP2 masks in response to RT might be dependent on the load used with increases in the rating of perceived effort and decreases in oxygen saturation when training to failure using lower but not higher loads [95]. However, there were no differences in total volume performed between mask or no mask conditions with any load. Therefore, although RT performance might not be particularly influenced by the use of masks, it should be suggested to train with higher loads and a lower number of repetitions to avoid discomfort [60].

Previous studies have suggested that SARS-CoV-2 might be transmitted by solid surfaces, where it might stay active for several days [96,97]. However, later evidence suggested that the risks are negligible [98]. Therefore, one should not be excessively worried about equipment sharing or cleaning, since regular hygiene practices might be enough to avoid transmission.

## 3. During: Resistance Training for People with COVID-19

COVID-19 involves an inflammatory response that affects different systems, including the neuromuscular system [99,100,101]. Its effects on muscle strength can be detected even in the absence of symptoms, with a strength loss of as much as 30% in 2 weeks of asymptomatic contamination [24]. COVID-19 patients under intensive care can lose 30% of the rectus femoris muscle cross-sectional area in the first 10 days [100], and 44% of them still have severely limited function for up to one month after weaning [102].

Previous studies have shown that reduced muscle strength is associated with physical inactivity in pulmonary patients [103] and is an important predictor of morbidity and mortality independent of the degree of respiratory limitation [104]. In agreement with this, muscle strength and mass are predictors of length of stay in patients with moderate to severe COVID-19 [105]. Consequently, it seems important to adopt strategies to maintain or increase muscle strength for all, from asymptomatic patients to those under intensive care, since exercise training during hospitalization due to acute respiratory conditions seems to be well tolerated and have infrequent adverse events [106,107,108]. In line with this, previous studies have suggested that rehabilitation programs starting within 30 days seem to bring the most benefits, as early exercise prevents neuromuscular complications; improves functional status in critical illness; and is considered effective, safe, and feasible [27,109].

In this regard, RT has been shown to promote benefits during pulmonary rehabilitation due to improvement in functional capacity, either performed alone or combined with aerobic training [110,111,112]. RT can be successfully performed as a stand-alone exercise strategy without increasing adverse events in chronic obstructive pulmonary disease patients under pulmonary rehabilitation [112].

This might seem counterintuitive, since a popular recommendation for periods of sickness is to rest. However, physical exercise might bring health benefits even in the case of virus contamination. Davies et al. [113] studied the effects of exercise on susceptibility to respiratory infection using a murine model. Mice received an intranasal challenge with the herpes simplex type 1 virus (HSV-1) and then were followed for three weeks under three situations: control, moderate short-term (30 min) exercise, and prolonged strenuous exercise to voluntary fatigue (2.5–3.5 h). The results showed that prolonged and strenuous exercise reduced the antiviral resistance of lung macrophages and increased both morbidity and mortality compared with either no exercise or short-term moderate exercise. However, in comparison to the control group, the groups that performed 30 min of moderate exercise showed a tendency to have a decreased morbidity (13 vs. 25%) and mortality (9 vs. 16%). Although human studies are lacking and assumptions based on animals should be made with great care, it is reasonable to suggest that RT sessions of short duration and a moderate intensity of effort could be recommended even in cases of infection, as previously suggested [60].

COVID-19 pathogenesis involves a delayed anti-viral response, which is followed by an excessive proinflammatory state [114]. The systemic inflammation is associated with disease severity, as shown by the higher serum levels of proinflammatory cytokine in the most affected patients [115]. This grants importance to interventions with anti-inflammatory properties, which is the case for RT [116,117,118]. It is also important to note that regulatory T lymphocyte (Treg) is associated with controlling inflammatory response and is reduced in severely ill patients [119,120], suggesting its important role in COVID-19 progression. In this regard, a previous study showed that RT can upregulate Treg [121] and that the regular practice of RT increases the levels of interleukin-10, an anti-inflammatory cytokine that is mainly produced by Treg cells [117,118].

Irisin might also have an important role in the benefits of RT for COVID-19 patients. A previous study indicated a positive effect of irisin on the expression of genes related to viral infection by SARS-CoV-2 [122]. Previous studies showed that exercise sessions acutely elevate irisin levels with higher increases in physically fit subjects [123] and that RT chronically elevates Irisin levels [124].

## 4. After: Resistance Training after COVID-19 Treatment

Respiratory diseases are associated with impairments in muscle function and the loss of lean body mass [125,126]. Survivors of severe acute respiratory diseases (SARS) might present a functional disability for as long as one year after discharge [127,128], and muscle wasting and weakness are frequent extrapulmonary conditions [128]. The manifestations include limb muscle weakness, muscle atrophy, and impairments in deep tendon reflexes [129].

COVID-19 patients show muscular disfunction similar to that observed in chronic obstructive pulmonary disease and IHD patients [130]. More than 80% of hospitalized COVID-19 patients, all without previous disability, showed reduced quadriceps muscle strength at discharge [131]. Studies carried out in COVID-19 patients who recovered from mild and moderate disease showed handgrip and quadricep weakness in 39.6% and 35.4% of the participants 12 weeks after discharge, respectively [132]. This might persist for a longer time, since persistent pulmonary function was impaired in up to 37% of the patients who suffered from SARS one year after discharge; their health status was also significantly decreased in comparison with healthy subjects [133,134], and exercise capacity was also remarkably lower than those found in the normal population for many months [134]. Moreover, patients admitted to intensive care units commonly present persistent dyspnea, anxiety, depression, impaired physical function, and a poor quality of life for up to 12 months after discharge [135,136,137]. Among these, physical function is one of the least likely to recover to normal values over the long term [28].

Previous studies have suggested that muscle strength should be regularly assessed after discharge [138,139], since the early diagnosis of sarcopenia appears to be of paramount importance in the management of post-acute COVID-19 patients [140]. The high prevalence of impairment in skeletal muscle strength and physical performance in patients recovering from COVID-19 suggests the need for rehabilitation programs after discharge. However, reduced muscle strength has been identified six months after discharge in one in six COVID-19 survivors, even when they were admitted to post-care facilities or received dietary counseling, physical activity guidance, or physiotherapy assistance [139]. Therefore, it is important to adopt specific strategies aimed at increasing muscle strength, such as working at adequate intensities and using blood flow restriction [140]. In this regard, previous studies have shown that a combination of aerobic training and RT increased muscle strength, functional capacity, and quality of life post COVID-19 infection [141,142,143].

It is important to consider possible risk factors when prescribing RT, as COVID-19 might be associated with cardiac complications that persist after discharge, especially arrhythmias, heart failure, myocardial injury, and increased risk of thromboembolism [144,145,146,147]. Nevertheless, RT has been shown to be safe and effective for several cardiac patients and has been recommended as a core component of cardiac rehabilitation for many decades [148,149,150]. However, it is important to consider proper program design to avoid complications, such as working with a lower number of repetitions, increasing rest intervals and reducing training volume [60].

The aforementioned evidence reinforces the importance of proposing a more oriented intervention to increase muscle function in COVID-19 survivors [60]; RT might be particularly interesting and has been shown to be safe and feasible in acute and chronic respiratory conditions [106,107,111,112].

## 5. Practical Considerations

When considering the period “before”, it is important to stimulate RT practice, as it improves general health and physical capacity. It should be highlighted that being physically active and having higher levels of muscle strength are associated with a lower risk of deleterious outcomes due to COVID-19. At this time, no specific care should be taken with RT design. However, special attention should be given to strategies for controlling contamination risk, such as ventilating rooms, social distancing, and considering the use of masks.

“During” contamination or in the presence of relevant symptoms, it is important to preserve the immune system while maintaining or increasing physical capacity. At this time, it is recommended to perform RT sessions for no longer than 30 min using a low number of repetitions (≤6), longer rests between sets (≥2 min), and submaximal effort (rate of perceive exertion ≤7). This recommendation can also help to reduce respiratory stress, particularly for participants returning to physical activity practices after new confinements due to coronavirus 2 variants (e.g., Delta, Omicron). Special attention should be given to the risk of contamination. Therefore, home-based training or training at places that allows for risk control are highly recommended. For patients under intensive care, the same recommendations can be adopted, with special attention given to the use of elastic bands, bodyweight exercises or exercise with no external load, as they can be performed inside the intensive care unit and even in the hospital bed.

RT should be performed immediately “after” recovery to increase functional capacity and allow the patient to return to their normal life and improve their quality of life. At this stage, RT should start progressing to traditional protocols. Training protocols should be tailored according to the clinical manifestations and considering the involvement of different symptoms. Our group have provided detailed recommendations in a previous article [60]; however, as a general rule, the recommendations for “during” can be maintained and progressed with clinical evaluation.

## 6. Conclusions

The COVID-19 pandemic brought us many important lessons about heath attention. So far, we have learned that the importance of physical exercise should be highlighted. The COVID-19 pandemic showed us that it is important to promote physical exercise and increase physical capacity to prevent complications in cases of contamination (before), maintaining health and controlling morbidities after infection (during), and helping to rehabilitate patients after discharge (after). In all these stages, RT might be a valuable tool based on both previous knowledge and the information obtained from the current pandemic. Among the characteristics that make RT a unique strategy, its adaptability for many spaces, equipment, and individual clinical characteristics can be highlighted. Therefore, it is recommended that health professionals and the general public be aware of its potential benefits during this period and in case of similar events in the future.

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
