# Peer review of "Resistance Training before, during, and after COVID-19 Infection: What Have We Learned So Far?"

_ijerph, 2022, doi:10.3390/ijerph19106323_

Round 1

Reviewer 1 Report

The authors did a nice job showing the relationship between muscle strength and COVID-19 hospitalization and risk of death. Based on that, they highlight the potential benefits of performing resistance training to prevent, treat, and recover from COVID-19 infection. Although, as state by the authors, a direct cause and effect relationship between resistance training in these different phases would lead to the proposed benefits, it is intuitive and reasonable to assume so. Authors provide some guidance on how resistance training should be tailored and highlight the important fact that it can be performed with space and equipment limitations. For all this, I congratulate the authors on this paper.

Below I have some minor points I would appreciate addressing.

Line 77. “isease” should read “disease”

Line 142: “dependent ON”

Line 150: “posterior evidence” should read “later evidence”, or “more recent evidence”

Line 156: “can be detected” in the past tense

Line 158: “can lose”, not can loose

Line 158: please check: “can loose 30% loss”? Does not make sense to me.

Lines 177-190: Authors should be cautions when citing animal studies. Please, reference human studies do address the topic discussed in this paragraph, or at the very least explicitly state human studies are lacking and assumptions based on animals should be made with great care.

Line 214: define acronyms at first entry

Finally, could authors provide any evidence that engaging in RT during or after COVID-19 infection leads to positive outcomes compared to remaining inactive? The papers cited were mostly on other diseases like COPD, pneumonia, etc. Thus, it would be very helpful and insightful to have data on COVID-19 as well. Please, double check the literature for that.

Author Response

Thank you very much for taking your time to review our article and for providing your valuable comments. We made our best to attend all comments and all the changes are highlighted in the text. Below we provide detailed answers.

Reviewer 1

The authors did a nice job showing the relationship between muscle strength and COVID-19 hospitalization and risk of death. Based on that, they highlight the potential benefits of performing resistance training to prevent, treat, and recover from COVID-19 infection. Although, as state by the authors, a direct cause and effect relationship between resistance training in these different phases would lead to the proposed benefits, it is intuitive and reasonable to assume so. Authors provide some guidance on how resistance training should be tailored and highlight the important fact that it can be performed with space and equipment limitations. For all this, I congratulate the authors on this paper.

***Thank you very much

Line 77. “isease” should read “disease”

Line 142: “dependent ON”

Line 150: “posterior evidence” should read “later evidence”, or “more recent evidence”

Line 156: “can be detected” in the past tense

Line 158: “can lose”, not can loose

Line 158: please check: “can loose 30% loss”? Does not make sense to me.

***Sorry, we have corrected the text.

Lines 177-190: Authors should be cautions when citing animal studies. Please, reference human studies do address the topic discussed in this paragraph, or at the very least explicitly state human studies are lacking and assumptions based on animals should be made with great care.

***We completely agree. Since we could not find human studies, we inserted the limitation.

Line 214: define acronyms at first entry

***Sorry. Since COPD only appears once, we decided to write the full name.

Finally, could authors provide any evidence that engaging in RT during or after COVID-19 infection leads to positive outcomes compared to remaining inactive? The papers cited were mostly on other diseases like COPD, pneumonia, etc. Thus, it would be very helpful and insightful to have data on COVID-19 as well. Please, double check the literature for that.

***We looked for “resistance training covid-19” on Pubmed on February 26 and found 149 results (https://pubmed.ncbi.nlm.nih.gov/?term=resistance+training+covid-19&format=abstract&sort=date&size=200 ). We carefully checked all titles and abstracts and found only one article, a review about concurrent training (Ahmadi Hekmatikar et al., 2022) that we have inserted.

Reviewer 2 Report

This is an actual and very comprehensive study in which authors have collected relevant and recent literature focusing on the benefits of RT during pandemic COVID-19.  The paper is generally well written and structured.

The information presented in the paper regarding the importance of regular physical activity highlighting the significance of RT during the current pandemic period are well established and proven with 148 publications . Accepting the paper type is a ‘Narrative review’ hereby, I have to mentioned that three quite similar ‘Review’ publications by the first Author are already available via search engines.  

Author Response

Thank you very much for taking your time to review our article and for providing your valuable comments. Indeed, some authors are involved with COVID-19 publications and this one is oriented for a more practical approach.

Reviewer 3 Report

The article is well written and easy to read. The subject matter is very topical and of particular interest in our area. Good work on the part of the authors.

Congratulations.

Author Response

Thank you very much!

Reviewer 4 Report

It is an interesting article that highlights the practice of resistance training before, during and after Covid-19 infection. The manuscript is well written, structured and referenced.

It is important that you specify the COVID-19 infection in the title.

Specific Comments:
Line 46, 47, 87... You should avoid using the first person. Please check this throughout your manuscript.

I consider that the title of point "1" should be eliminated, and call it an introduction, without numbering.

The phrase between lines 93 and 95 must be accompanied by references.

Line 207. Check the wording of the title.

Author Response

Thank you very much for taking your time to review our article and for providing your valuable comments. We made our best to attend all comments and all the changes are highlighted in the text. Below we provide detailed answers.

It is an interesting article that highlights the practice of resistance training before, during and after Covid-19 infection. The manuscript is well written, structured and referenced.

It is important that you specify the COVID-19 infection in the title.

***Great suggestion. We have inserted “infection” in the title

Specific Comments:

Line 46, 47, 87... You should avoid using the first person. Please check this throughout your manuscript.

***Sorry, we have reviewed the text.

I consider that the title of point "1" should be eliminated, and call it an introduction, without numbering.

***Thank you, we have changed the text.

The phrase between lines 93 and 95 must be accompanied by references.

***Sure! We have inserted the references

Line 207. Check the wording of the title.

***Thank you, we have changed the title

Reviewer 5 Report

General Comments.

The authors provide an understandable article and are well referenced. They provide a general view about the exercise role over the COVID-19 pandemic when the restrictions were needed to avoid a higher spread of the virus. They submitted a paper with a strong argumentation about the context and physiological response in resistance training, but specific practical recommendations are light. Therefore this article could be of little utility for health professionals and trainers. Restrictions influenced the lifestyle of citizens, and many coaches had to adjust the training programmes in the new context. So, I suggest the authors include alternative methods of training. Please note that my comments are just a suggestion for improving professional usefulness. See my specific comments below.

Specific Comments.

Line 58. Double spaced before reference [3,4].

Line 68. Double spaced after reference [11, 15].

Line 91. The title of this section could lead to understanding that exercise is a treatment for avoiding infection. I suggest changing to “Before: Considerations for preventing COVID-19 complication during resistance training practice”.

Line 95. Double spaced after reference [11, 15].

Line 109-111. Authors could provide a range of time or repetitions for volume parameter, percentage of one-rep max or similar relative intensity which avoid metabolic stress. Even, authors could provide a kind of training like example (plyometric, isometric, and so on)

Line 120-122. Authors could indicate how training is sufficient to achieve a similar effect to traditional resistance training. Also, the authors could include plyometrics and isometric training recommendations.

Line 126. Double spaced after reference [79, 81, 82].

Line 219. Double spaced after reference [131].

Line 236-238. Authors could give an example of a kind of training like blood flow restriction with low-intensity resistance training. According to Hughes (2017), this procedure has shown a positive effect against sarcopenia after a hospital stay.

Hughes L, Paton B, Rosenblatt B, et al. Br J Sports Med 2017;51:1003–1011.

Line 261-263. Concerning training intensity, “submaximal effort” is a general recommendation. Could the authors provide more specific details? I mean the metrics to monitor the intensity like RPE or %RM.

Author Response

Thank you very much for taking your time to review our article and for providing your valuable comments. We made our best to attend all comments and all the changes are highlighted in the text. Below we provide detailed answers.

General Comments.

The authors provide an understandable article and are well referenced. They provide a general view about the exercise role over the COVID-19 pandemic when the restrictions were needed to avoid a higher spread of the virus. They submitted a paper with a strong argumentation about the context and physiological response in resistance training, but specific practical recommendations are light. Therefore this article could be of little utility for health professionals and trainers. Restrictions influenced the lifestyle of citizens, and many coaches had to adjust the training programmes in the new context. So, I suggest the authors include alternative methods of training. Please note that my comments are just a suggestion for improving professional usefulness. See my specific comments below.

***Thank you very much

Specific Comments.

Line 58. Double spaced before reference [3,4].

Line 68. Double spaced after reference [11, 15].

Line 91. The title of this section could lead to understanding that exercise is a treatment for avoiding infection. I suggest changing to “Before: Considerations for preventing COVID-19 complication during resistance training practice”.

Line 95. Double spaced after reference [11, 15].

***Sorry, we have amended the text accordingly

Line 109-111. Authors could provide a range of time or repetitions for volume parameter, percentage of one-rep max or similar relative intensity which avoid metabolic stress. Even, authors could provide a kind of training like example (plyometric, isometric, and so on)

***Great suggestion. We have clarified the recommendations.

Line 120-122. Authors could indicate how training is sufficient to achieve a similar effect to traditional resistance training. Also, the authors could include plyometrics and isometric training recommendations.

***we have inserted plyometric training. However, although isometric training is a very interesting approach it might also involve the use of machines and equipment; therefore, we think that it is contemplated when we site elastic bands and training with no load.

Line 126. Double spaced after reference [79, 81, 82].

Line 219. Double spaced after reference [131].

*** sorry, we have corrected the text.

Line 236-238. Authors could give an example of a kind of training like blood flow restriction with low-intensity resistance training. According to Hughes (2017), this procedure has shown a positive effect against sarcopenia after a hospital stay.

Hughes L, Paton B, Rosenblatt B, et al. Br J Sports Med 2017;51:1003–1011.

***Thank you, we have inserted the information

Line 261-263. Concerning training intensity, “submaximal effort” is a general recommendation. Could the authors provide more specific details? I mean the metrics to monitor the intensity like RPE or %RM.

***Sorry, we provide more details